# Pleiotropic Functions of *FoxN1*: Regulating Different Target Genes during Embryogenesis and Nymph Molting in the Brown Planthopper

**DOI:** 10.3390/ijms21124222

**Published:** 2020-06-13

**Authors:** Yu-Xuan Ye, Chuan-Xi Zhang

**Affiliations:** 1Institute of Insect Science, Zhejiang University, Hangzhou 310058, China; yeyuxuan@zju.edu.cn; 2State Key Laboratory for Managing Biotic and Chemical Threats to the Quality and Safety of Agro-Products, Institute of Plant Virology, Ningbo University, Ningbo 315211, China

**Keywords:** *FoxN1*, brown planthopper, pleiotropism, embryogenesis, keratin gene

## Abstract

*FoxN1* gene belongs to the forkhead box gene family that comprises a diverse group of “winged helix” transcription factors that have been implicated in a variety of biochemical and cellular processes. In the brown planthopper (BPH), *FoxN1* is highly expressed in the ovaries and newly laid eggs, where it acted as an indispensable gene through its molecular targets to regulate early embryonic development. Moreover, the results of the RNAi experiments indicated that *Nilaparvata lugens*
*FoxN1* (*NlFoxN1*) exhibited pleiotropism: they not only affected the embryogenesis, but also played an important role in molting. RNA-seq and RNAi were further used to reveal potential target genes of *NlFoxN1* in different stages. In the eggs, ten downregulated genes were defined as potential target genes of *NlFoxN1* because of the similar expression patterns and functions with *NlFoxN1*. Knockdown of *NlFoxN1* or any of these genes prevented the development of the eggs, resulting in a zero hatchability. In the nymphs, *NlFoxN1* regulated the expression of a keratin gene, type I cytoskeletal keratin 9 (*NlKrt9*), to participate in the formation of an intermediate filament framework. Depletion of *NlFoxN1* or *NlKrt9* in nymphs, BPHs failed to shed their old cuticle during nymph-to-nymph or nymph-to-adult molting and the mortality was almost 100%. Altogether, the pleiotropic roles of *NlFoxN1* during embryogenesis and nymph molting were supported by the ability to coordinate the temporal and spatial gene expression of their target genes.

## 1. Introduction

*FoxN1* gene belongs to the forkhead-box (Fox) gene family that encodes a large family of transcription factors characterized by a “winged-helix” DNA-binding domain [1]. Forkhead domain (FHD) is very well conserved across the Fox family and across various eukaryotic species; it extends about 100 amino acids in length [2].

*FoxN1* was first well characterized both in mouse and humans, and designated as a nude gene [3]. FoxN1 is a transcription factor specific to epithelial cells and known to regulate the differentiation of several tissues [4]. *FoxN1* expression could be detectable in the entire skin, particularly interfollicular epidermis and hair follicle, including developing whisker pads, nail primordial, hair follicles of eyebrows, and the epidermis of mouth, nose, ears, and tail [5]. In the skin, *FoxN1* is mainly expressed in the keratinocytes of the first suprabasal layer and is a key transcription factor involved in the regulation of keratinocytes growth and differentiation [6,7]. The lack of functional FoxN1 protein leads to a nude phenotype in mice and humans, which is characterized by the lack of visible hair, and skin and nail abnormalities [8]. *FoxN1* gene expression in the skin, as also seen in the thymus [9], mutations in the “nude” *FoxN1* gene induce the hairless phenotype, associated with a severe combined immunodeficiency (SCID) phenotype [10]. During embryogenesis, *FoxN1* is essential for the development of the thymus, the primary lymphoid organ that supports T-cell development and selection [11]. Similarly to the mouse *Foxn1* gene, the human *FOXN1* gene contains eight coding exons and two different first exons (not contributing to the protein sequence) that undergo alternative splicing [7]. Exon 1a and exon 1b are under the control of two different promoters, thereby affording *FOXN1* tissue specificity; the promoter 1a is active both in thymus and skin and the promoter 1b only in the skin [12].

The pleiotropic roles of Fox proteins during the embryonic development and homeostasis of adult tissues is supported by the ability to coordinate the temporal and spatial gene expression of their target genes [13]. However, the molecular targets regulated by FoxN1 in different tissues or developmental stages to exert different functions remain to be elucidated.

Because of the absence of thymus organs, the expression and function of *FoxN1* in insects might be different from those in mammals. To better understand the pleiotropic mechanism of *FoxN1* and its potential application in insect pest control, we used the brown planthopper (BPH), *Nilaparvata lugens* Stål (Hemiptera: Delphacidae), one of the most destructive insect pests of rice crops [14], as a model to explore the functions and the potential transcriptional targets of *FoxN1* in insects.

## 2. Results

### 2.1. Sequence Analysis and Expression Patterns of NlFoxN1

There are two duplicated *FoxN1* genes in the *N. lugens* genome (Figure 1a). *NlFoxN1a* and *NlFoxN1b* share the same number of exons, the same cDNA sequences, and differ only in intron length. Similarly to the mouse and human *FoxN1* genes, *NlFoxN1* contains eight coding exons [12]. The cDNA of *NlFoxN1* was amplified, cloned, and sequenced. The complete ORF sequence of *NlFoxN1* is 1941 bp long. BLAST analysis of the 111-amino-acid DNA binding domain of *Nl*FoxN1 in NCBI revealed its homology with *Papilio xuthus* Forkhead box protein N1 with a percentage of identity of 75%. Multiple sequence alignments were performed among FoxN1 orthologs from 11 species using the ClustalX program, and the result showed that the sequences of FoxN1 were highly conserved in vertebrates and invertebrates (Figure 1b).

### 2.2. Temporal and Spatial Expression Patterns of NlFoxN1

The spatial and temporal expression patterns of *NlFoxN1* were assessed by real-time quantitative PCR (qPCR). To explore the developmental expression patterns, total RNA was extracted from various developmental samples containing all life stages of BPHs (eggs, nymphs in five different instars, female adults, and male adults). The results showed that *NlFoxN1* was highly expressed in newly laid eggs, which was the early stage of embryogenesis, indicating that *NlFoxN1* might play an important role in the embryonic development of fertilized eggs. Compared to that in males, the transcripts of *NlFoxN1* in females were maintained at a relatively higher level (Figure 2a). The tissue samples used for the tissue-specific expression patterns were extracted from random adults including gut, salivary gland, fat body, integument, testis, and ovary. *NlFoxN1* was mainly expressed in the ovaries, suggesting that eggs have begun to express *NlFoxN1* in the maternal stage (Figure 2b).

### 2.3. NlFoxN1 Was Indispensable for Embryogenesis

To determine the function of the *NlFoxN1*, we conducted RNAi experiments. To avoid off-target effects, RNAi experiments were replicated by choosing two non-overlapping regions as targets. The qPCR analysis revealed that each dsRNA efficiently suppressed the transcript levels of their target genes (Appendix A).

Since *NlFoxN1* was highly expressed in the eggs, we conducted the RNAi experiments on newly emerged female adults, at the time when the ovaries were not fully developed, to observe the phenotypes of egg production and embryonic development in the next generation. The ovaries developed normally after injection, and there was no significant difference in egg production compared with the ds*GFP*-treated group, but the hatchability of the eggs was seriously affected (Figure 3a). The eggs without expression of *NlFoxN1* could not develop normally. The red eyespots (compound eye buds) that should appear about 4 days after egg production did not appear, without exception, indicating that embryonic development is terminated before the stage of embryonic movement (Figure 3b).

### 2.4. Potential Targets of NlFoxN1 in the Eggs

Large scale RNAi and RNA-seq were used to reveal potential target genes of *Nl*FoxN1 in the eggs. A total of 63 differentially expressed genes (DEGs) were identified by transcriptome sequencing, among which, 59 genes were significantly downregulated after knockdown of *NlFoxN1* in the newly laid eggs. qPCR assays confirmed that 23 of the DEGs were downregulated by *NlFoxN1* including several transcription factors, and the four upregulated genes were proven insignificant (Figure 3c). To extrapolate the biological processes of the DEGs, we performed Gene Ontology (GO) term enrichment analysis. We found that a total of 82.6% of the 23 downregulated genes were significantly enriched in binding activities (GO:0005488), which is defined as interaction of a molecule with one or more specific sites on another molecule, including protein and nucleic acid binding activities (Figure 3d).

To determine the functions of the 23 downregulated genes, dsRNAs targeting each of these genes were injected into newly emerged female adults (within two hours), respectively. The RNAi results indicated that ten genes were indispensable to embryogenesis and knockdown of any of these ten genes would therefore result in a zero-hatching rate (Figure 3e). Furthermore, these ten genes had similar expression patterns to *NlFoxN1*, highly expressed in the newly laid eggs (Figure 3f). These results suggested that these ten genes were the potential target genes of *NlFoxN1*, and therefore accounted for the lethal effect of *NlFoxN1* knockdown in early egg stages.

### 2.5. NlFoxN1 Regulated Keratin Genes to Affect Molting Process

In addition to the high expression levels in eggs and adults, *NlFoxN1* also had a certain level of expression in nymphs. To investigate the possible functions of *NlFoxN1* in nymphs, dsRNA was injected into newly emerged 4th and 5th instar nymphs (within two hours), respectively.

Injection of dsRNA for *NlFoxN1* in early fifth-instar nymphs led to a lethal phenotype, the death rate reached almost 100% (Figure 4a). All dead BPHs exhibited the same phenotype, they failed to shed their old cuticles and died quickly during nymph-to-adult molting (Figure 4b). The same phenotype was found between 4th to 5th nymph molting after injecting *NlFoxN1* dsRNA into early fourth-instar nymphs (Figure 4c). To investigate the changes in gene expression resulting from depletion of *NlFoxN1*, we made transcriptomic analysis using RNA-seq. A total of 248 up- and 652 downregulated genes were detected in the *NlFoxN1*-RNAi nymphs (4th 48 h). The GO terms of the DEGs were significantly enriched in the chitin-based cuticle (GO:0005214) for both biological process and molecular function categories, while for cellular components, the enriched GO terms were related to the extracellular region (GO:0005576) (Figure 5).

Among these DEGs, three keratin genes encoding type I cytoskeletal keratin 9 (Krt9), type I cytoskeletal keratin 10 (Krt10), and type II cytoskeletal keratin 1 (Krt1) were downregulated by ds*NlFoxN1*. The keratin genes were highly expressed in the integument (cuticles) (Figure 6a). The qPCR results further confirmed that the three keratin genes were significantly downregulated after the injection of ds*NlFoxN1* in 4th instar nymphs (Figure 6b). To determine the relationship between the keratin genes and the high mortality, microinjection experiments were performed with dsRNAs targeting *NlKrt1*, *NlKrt9*, and *NlKrt10*, respectively. The RNAi results showed that knockdown of *NlKrt9* resulted in the same phenotypes with *NlFoxN1*. The *NlKrt9*-RNAi nymphs had a high mortality (Figure 6c) and molting dysfunction (Figure 6d). However, injection of the dsRNA for *NlKrt1* or *NlKrt10* did not result in any abnormal molting or other observable morphological or inter-structural abnormalities.

## 3. Discussion

The results of the RNAi experiments indicated that *NlFoxN1* exhibited pleiotropism; not only affecting the egg hatching rate in eggs, but also playing an important role in molting in nymphs. RNA-seq has become a powerful tool to investigate transcriptome profiling using deep-sequencing technologies [15]. To investigate DEGs of *NlFoxN1* in different stages, we used RNA-seq to profile the transcriptome after RNAi in eggs and nymphs, respectively. The DEGs verified by qPCR and RNAi assays were defined as the potential target genes.

*NlFoxN1* was highly expressed in the very beginning of the eggs and was indispensable to embryonic development, probably acting as a “pioneer” factor which is involved in the regulation of cell differentiation in the early stages of embryonic development [16]. Pioneer factors are able to access their target sequence by interacting with condensed chromatin and open the way for the expression of formerly silent important transcription factors, thus making the regulated genes available for activation [17]. Such activity might explain that potential target genes of *NlFoxN1* in the newly laid eggs contained multiple transcription factors. *Nl*FoxN1 activated the ten potential target genes which had the similar expression patterns and functions with *NlFoxN1* to participate in embryogenesis in eggs.

In nymphs, cuticles are a complex composite material, made of chitin filaments embedded in cuticular proteins [18]. It provides structural and mechanical support by serving functionally as both skin and skeleton [19]. There are obvious keratinization defects in the epidermis and nails of nude mice, and *Foxn1* malfunction can result in the suppression of keratin genes [20]. Our RNA-seq results indicated that three *N. lugens* keratin genes (*NlKrt1*, *NlKrt9,* and *NlKrt10*) were downstream genes of *NlFoxN1*. Moreover, knockdown of *NlKrt9* results in the same phenotypes with that of *NlFoxN1*-RNAi; defects in molting with a high mortality. *Krt9* is an intermediate filament (IF) protein, specifically expressed in the suprabasal layers of the palmoplantar epidermis in humans [21]. Dominant-negative mutations in the *Krt9* gene cause epidermolytic palmoplantar keratoderma, a rare skin disorder [22]. The IF framework that exists in insect cells is similar to the IF framework that exists in mammalian cells [23]. The evidence strongly suggests that *Nl*FoxN1 participates in the formation of the IF framework by regulating the expression of *NlKrt9,* thereby influencing the molting process.

The pleiotropism of *FoxN1* in the mammals is controlled by alternative splicing with two different first exons [12]. *NlFoxN1* had only one splicing but two duplicated genes in the *N. lugens* genome, indicating that the pleiotropic mechanism of *NlFoxN1* was different from that in the mammals. However, whether the pleiotropic mechanism of *NlFoxN1* is regulated by duplicated genes requires further study.

The evolutionary conservation of the crucial DNA-binding domain between orthologous members of the Fox gene family is remarkable [24]. Our analysis showed that the sequences of FoxN1 are highly conserved both in vertebrates and invertebrates. The *Drosophila* homologue of *FoxN1* is *jumeaux* [25], which is required for normal eye and wing morphogenesis and appears to act in the modification of chromatin structure [26]. In *Aedes aegypti*, *FoxN1* is strongly expressed in the ovaries and is important in regulating mosquito reproduction [27]. This means that the roles of *FoxN1* in embryogenesis and cuticles also seem somewhat conserved in insects.

Altogether, *NlFoxN1* regulated ten potential target genes including several transcription factors to initiate embryogenesis in the eggs, and regulated keratin genes to maintain the homeostasis of nymph cuticles, thus demonstrating the pleiotropic functions of *NlFoxN1*. These findings may furthermore stimulate the design and development of novel insecticides which could control both current and next generation populations.

## 4. Materials and Methods

### 4.1. Insects

The insects used in this study were obtained from local rice fields at Zhejiang University, Hangzhou, Zhejiang, China. The insects were reared on fresh rice seedlings (Xiushui 134) at 25 ± 1 °C and 60–70% relative humidity under a 16 h/8 h (light/dark) photoperiod [28].

### 4.2. Gene Identification and Sequence Analyses

*Nilaparvata lugens* genomic [29] and transcriptomic databases were screened for the gene encoding FoxN1 against the amino acid sequences from *Drosophila melanogaster*, *Mus musculus*, and *Homo sapiens*, which were obtained from GenBank. The full-length cDNA sequence was obtained from transcriptomic databases and confirmed by RT-PCR. The gene structure display server program (GSDS2.0) was employed to identify gene structures [30]. Multiple sequence alignments were performed using the ClustalX program [31]. The primers used here are shown in Appendix A.

### 4.3. Expression Pattern Analysis

Total RNA from whole insects at various developmental stages or tissue samples was isolated using a TRIzol Total RNA Isolation Kit (Takara, Kyoto, Japan). Developmental samples were collected from different stages of BPHs (*n* = 15–20), including 8 egg samples, 30 nymph samples, 7 female adult samples, and 4 male adult samples. Similarly, various tissue samples, including integument, gut, fat body, salivary gland, testis, and ovary were dissected from random female adults 48–72 h after adult emergence. Three biological replications were performed for each developmental and tissue sample. Each biological replicate consisted of *N. lugens* eggs, nymphs, adults or specific tissue samples. To investigate the developmental and tissue-specific expression patterns, qPCR was conducted using ChamQ SYBR Color qPCR Master Mix (Vazyme Biotech Co.,Ltd). The pairs of gene-specific primers were designed using the Primer Premier 6 program (Appendix A). The *N. lugens* housekeeping gene for 18S ribosomal RNA (*Nl18S*) (GenBank accessionnumber JN662398.1) was used as an internal control. 

### 4.4. RNAi Effects on N. lugens

The double-stranded RNA (dsRNA) was synthesized using a T7 High Yield RNA Transcription Kit (Vazyme Biotech Co.,Ltd). A unique region of each gene was chosen as a template for dsRNA synthesis. The primers used for the dsRNA synthesis can be found in Appendix A. The dsRNA for *GFP* was used as a negative control for the nonspecific effects of dsRNA. Microinjection of planthoppers with dsRNA was carried out according to a previously reported method [32]. One hundred and fifty insects were used for each gene treatment, and each treatment was conducted in triplicate. Each insect was injected with 10 μL of dsRNA, at a concentration of 5 ng/μL. Samples were collected from a set of 6–10 insects to evaluate the RNAi effects of each gene. A second, non-overlapping region was selected for dsRNA synthesis to overcome possible off-target effects.

### 4.5. Differential Expression Analysis Using RNA-seq

The newly emerged female adults were injected with dsRNA, paired with normal male adults, and kept for three days on fresh rice seedlings until they reached sexual maturity. Approximately 500 eggs newly laid on rice stems were carefully dissected for total RNA extraction. Thirty individuals, 48 h after injecting dsRNA in the fourth instar nymphs, were homogenized for total RNA extraction. The dsRNA for *GFP* was used as a negative control for the nonspecific effects of the dsRNA. Each treatment involved three sets of biological replicates. The cDNA library preparation and Illumina sequencing were performed using Annoroad (Beijing, China).

The clean reads were aligned to the reference genome using HISAT2 [33]. The low-quality alignments were filtered with SAMtools [34]. TPM expression values were calculated using featureCounts for genes [35]. Differential expression analysis was carried out using the DESeq2 package [36]. The differentially expressed genes needed to meet the following conditions: false discovery rate (FDR) < 0.05 and absolute value of the log2 ratio > 2.

## Figures and Tables

**Figure 1 ijms-21-04222-f001:**
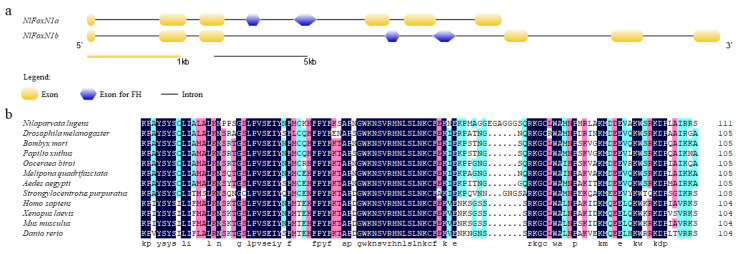
Sequence analysis of *NlFoxN1*. (**a**) Exon-intron analysis of *NlFoxN1* was carried out by GSDS2.0. (**b**) Alignment of amino acid sequences of forkhead domain (FHD) from 12 species was carried out by DNAMAN.

**Figure 2 ijms-21-04222-f002:**
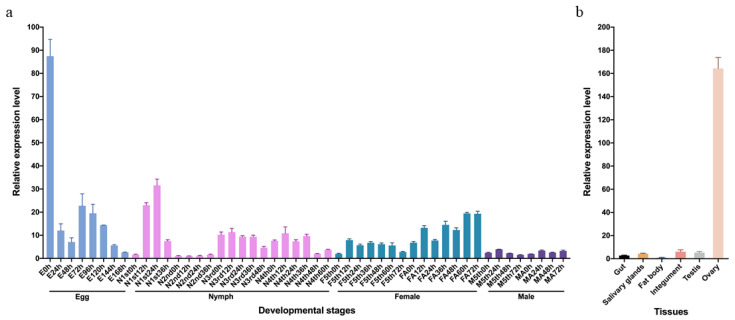
Temporal and spatial expression patterns of *NlFoxN1*. (**a**) Expression patterns in different development stages. E: egg; N: nymphs regardless of gender; F: female; FA: female adult; M: male; MA: male adult. (**b**) Expression patterns in different tissues. Developmental samples were extracted from whole insects at all life stages of BPHs (*n* = 200 eggs, *n* = 30 nymphs and adults). Samples were collected every 12 h or 24 h from the very beginning of each stage. Tissue samples were dissected from 50 random adults 48–72 h after eclosion. *Nl18S* was used as an internal control gene for qPCR. Mean ± standard error of the mean (s.e.m.) from three experiments.

**Figure 3 ijms-21-04222-f003:**
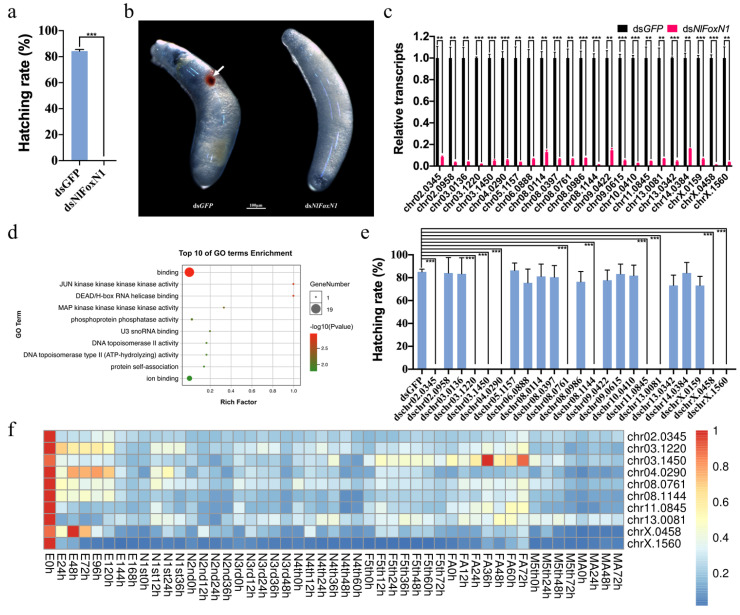
*NlFoxN1* was indispensable for embryogenesis. (**a**) The hatching rate of the eggs after knockdown of *NlFoxN1*. (**b**) The lethal phenotypes of eggs after knockdown of *NlFoxN1*. The red eyespot was marked by the white arrow. (**c**) qPCR verification of the differentially expressed genes (DEGs). (**d**) Gene Ontology (GO) term enrichment analysis of the 23 verified DEGs in eggs. The GO enrichment study was performed using OmicShare tools. (**e**) The hatching rates of the eggs after knockdown of the 23 verified DEGs, respectively. (**f**) The temporal expression patterns of the ten potential targets. Heatmap was plotted using pheatmap R package based on the qPCR data from three biological replicates. dsRNA (50 ng per insect; *n* = 100) was injected into newly emerged female adults (within two hours). ds*GFP* was injected as a negative control for the nonspecific effects of dsRNA. Mean ± s.e.m. from three experiments. ** *p* < 0.01 and *** *p* < 0.001 (Student’s *t*-test), difference from ds*GFP*.

**Figure 4 ijms-21-04222-f004:**
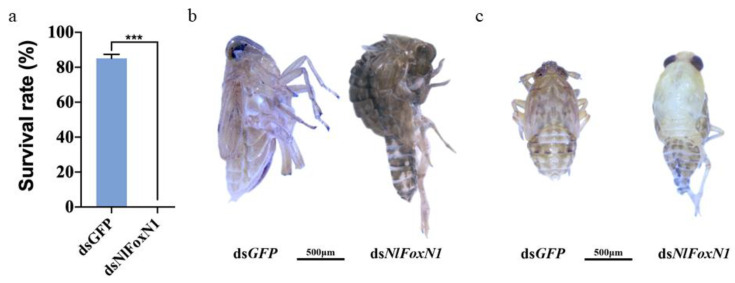
Knockdown of *NlFoxN1* in nymphs affected the molting process. (**a**) The survival rate after *NlFoxN1*-RNAi in the fifth instar. (**b,c**) The lethal phenotypes after injection of dsRNA for *NlFoxN1* in the fifth instar (**b**) and the fourth instar (**c**). dsRNA (50 ng per insect; *n* = 100) was injected into BPHs at the very beginning of each instar. ds*GFP* was injected as a negative control for the nonspecific effects of dsRNA. Mean ± s.e.m. from three experiments. *** *p* < 0.001 (Student’s *t*-test), difference from ds*GFP*.

**Figure 5 ijms-21-04222-f005:**
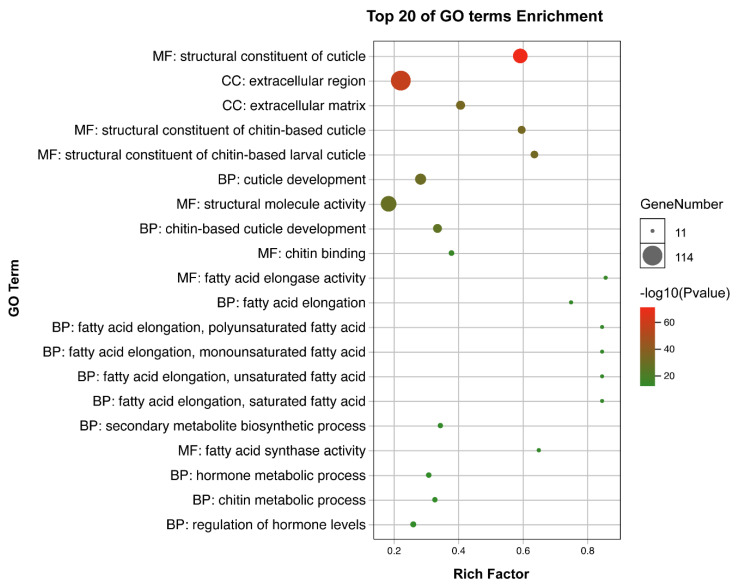
The GO enrichment analysis of the DEGs in nymphs. The GO enrichment study was performed using OmicShare tools. MF: molecular function, CC: cellular component, bp: biological process.

**Figure 6 ijms-21-04222-f006:**
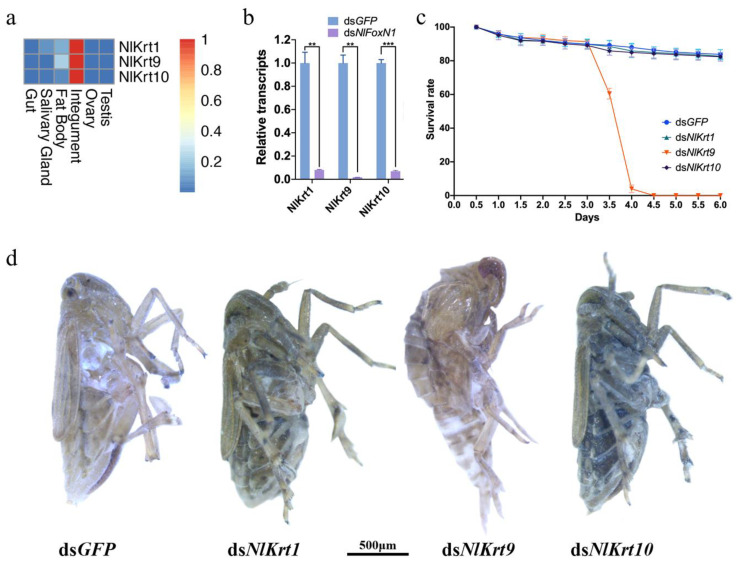
*NlFoxN1* regulated keratin genes to affect molting process. (**a**) Tissue-specific expression patterns of the keratin genes. (**b**) Knockdown of *NlFoxN1* decreased expression levels of keratin genes in the nymphs. (**c**) The survival rates after knockdown of the keratin genes in the fifth instar. (**d**) The phenotypes after injection of dsRNA for keratin genes in the fifth instar. dsRNA (50 ng per insect; *n* = 100) was injected into BPHs at the very beginning of the fifth instar. ds*GFP* was injected as negative control for the nonspecific effects of dsRNA. Mean ± s.e.m. from three experiments. ** *p* < 0.01 and *** *p* < 0.001 (Student’s *t*-test), difference from ds*GFP*.

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
