# Peer review of "Pleiotropic Functions of FoxN1: Regulating Different Target Genes during Embryogenesis and Nymph Molting in the Brown Planthopper"

_ijms, 2020, doi:10.3390/ijms21124222_

Round 1

Reviewer 1 Report

This is a well written manuscript that provides new information on function of genes/ transcriptional factors from the the forkhead family. However, I would like to see the authors to expand the discussion section by talking about evolutionary as well as applied significance of their findings in additional 1-2 paragraphs. Since not much is known about the genes of interests in insects and/or brown planthopper, authors could also discuss the putative function of different genes in other insect or eukaryotic species.

In the methods section, authors talk about expression pattern analysis, but do not describe the qPCR details. This information needs to be included in the methods section. Also, authors have not clearly described replication strategy used for the gene expression work. From the description that is provided it appears that authors have not replicated qPCR experiments, which is a fatal flaw in experimental design. Lastly, authors need to explain what they mean by digital gene expression analysis in the methods section along with appropriate references.

Overall this manuscript can be reconsidered for publication after the authors address abovementioned major issues.

Minor comments:

  1. L32: Change 'eukaryote' to 'eukaryotic'.
  2. 90: Change 'from' to 'in'.
  3. L98-99: Information provided in this sentence is appropriate for the methods section or the discussion section. Please move these sentences to an appropriate section.
  4. L103-104: Did you choose any upregulated genes for qPCR analysis?
  5. L109: Replace comma with 'and'.

Reviewer 2 Report

The manuscript titled “Pleiotropic functions of FoxN1: Regulating different target genes during embryogenesis and nymph molting in the brown planthopper” by Ye and Zhang reports the function of FoxN1 in N. lugens through gene expression profiling by deep sequencing and knock-down studies. These works identified the important roles of NlFoxN1 in egg and nymph development and the potential target genes of NlFoxN1 through which NlFoxN1 might function. The findings of the present study are interesting and novel. In spite, some critical data need to be supplemented and the organization of the manuscript needs to be improved before consideration of publication. I suggest the authors to address the below:

Major comments:

Summarize and generate plots of 63 DEGs from the GO analysis and include a figure(s) from the analysis in the main text in addition to the current Table S2.

In line 105–106, the authors states “we found that a total of 82.6% of the downregulated genes were significantly enriched in binding…”  The term “binding” is less specific and clear.  Please define or explain it further. In addition, what biological pathway is the “binding” implicated with? DNA metabolism? Transcription? Please elaborate it in the text.

The legend of Fig. 2 should include more detailed description to avoid confusion. It lacks the description that the figures include the 10 DEGs upon NlFoxN1 depletion.

Could the KD validation/efficiency data for the 10 DEGs be included such as RT-PCR? In addition, please compose a new figure 3 with the DEGs data including the GO plot, 10 DEG KD validation, and the hatching rate from the KD experiment with the 10, as well as the current figure 3 (egg phenotype as Fig. 3a).  

Figs. 4 and 5 should be combined as Fig. 4 because they are for the same context and conclusions. In addition, in Fig. 4 description in the main text (ines 119–120), keratin genes don’t yet appear at this point. Therefore, the keratin gene data should not be included in Fig. 4. Authors need to organize the figures according to the main text neatly so that the readers don’t have to go up and down to find correct figures unnecessarily.

The authors should present a summarizing plot(s) from the GO analysis with NlFoxN1-KD in nymphs. This perhaps can be Fig. 5.

NlFoxN1 doesn’t appear to be expressed much except for the very early egg period. Do you think that the protein level could differently fluctuate from the RNA level?  

The keratin gene results can be combined to be displayed in Fig. 6 or combined with Fig. 5. They need to be separately displayed from the current Fig. 4.  

Current Fig. 4 lacks error bars and statistical validations.

Minor comments:

When first appearing, state what the acronyms stand for, e.g. N. lugens FoxN1 (NlFoxN1).

In Fig. 1a, the black solid lines should be straight. The vent lines make it difficult to compare with the 5 kb scale bar which is straight.

In Fig. 3, indicate all known regions in the control egg (dsGFP) such as “red eyespots.”

Could the dsNlKrt1 or 10 phenotypes be shown along with dsNlKrt9 for comparisons?

In the introduction, could what is known about FoxN1 in drosophila be incorporated?

Round 2

Reviewer 1 Report

Overall comments:

Authors have not clearly explained the changes that they made to the manuscript in their cover letter and the changes that were made in addition to the suggested changes (e.g., including new data in Fig. 3) have not been shown with track changes.  Hence I cannot review or make a decision on acceptance of this manuscript at this time. Be sure to include a "red-line" version for third round of review of this manuscript.

Specific comments:

First, authors need to include description on biological replication (if any) in methods section 4.3. If they did not perform any biological replicates they should explicitly say so in the methods section.

Second, provide rationale for including new data and Fig. 3 and switching RNA-seq data with qPCR data in Fig. 2. It seems a bit perplexing and unethical to just swap data-sets when fatal flaws in the design are pointed out.

Reviewer 2 Report

I see that the manuscript has been improved according to my previous comments. Some minor comments are:

Figure 1b- Use lighter color than black for the shade with black letters so that the letters can be seen.

Reduce the size of Figure 5 as it appears too large, compared to other figures. 

Round 3

Reviewer 1 Report

I appreciate the authors providing a redline version of the manuscript and clarifying that some of the revisions were made in response to another reviewer's comments. The other reviewers comments were not shared with me during round 2 of review, but I see them now.

Also, I thank the authors for providing clarification on the important topic of biological replication. Since your paper will be used by other researchers to perform qPCR and other experiments, I think it is important to mention the biological replication requirement. I just have a few important changes on how biological replication sentence is mentioned in section 4.3.

Delete the following (lines 248-249): Each time point or tissue 248 was carried out for three biological replicates.

Add the following sentences to L245 right after..."adult development" and before.... "To investigate the...": "Three biological replications were performed for each developmental and tissue sample. Each biological replicate consisted on xx–xx eggs, nymphs, adults or specific tissue samples".

After authors make the above mentioned important changes to section 4.3, the paper can be recommended for publication.
